# Polymeric Nanoparticles in Brain Cancer Therapy: A Review of Current Approaches

**DOI:** 10.3390/polym14142963

**Published:** 2022-07-21

**Authors:** Chad A. Caraway, Hallie Gaitsch, Elizabeth E. Wicks, Anita Kalluri, Navya Kunadi, Betty M. Tyler

**Affiliations:** 1Hunterian Neurosurgical Research Laboratory, Department of Neurosurgery, Johns Hopkins University School of Medicine, Baltimore, MD 21205, USA; ccarawa2@jhmi.edu (C.A.C.); hgaitsc1@jhmi.edu (H.G.); ewicks2@jhmi.edu (E.E.W.); akallur1@jhmi.edu (A.K.); navyakunadi@gmail.com (N.K.); 2NIH-Oxford-Cambridge Scholars Program, Wellcome—MRC Cambridge Stem Cell Institute and Department of Clinical Neurosciences, University of Cambridge, Cambridge CB2 1TN, UK; 3University of Mississippi School of Medicine, University of Mississippi Medical Center, Jackson, MS 39216, USA

**Keywords:** brain, cancer therapy, drug delivery, glioblastoma multiforme, nanoparticles, polymers

## Abstract

Translation of novel therapies for brain cancer into clinical practice is of the utmost importance as primary brain tumors are responsible for more than 200,000 deaths worldwide each year. While many research efforts have been aimed at improving survival rates over the years, prognosis for patients with glioblastoma and other primary brain tumors remains poor. Safely delivering chemotherapeutic drugs and other anti-cancer compounds across the blood–brain barrier and directly to tumor cells is perhaps the greatest challenge in treating brain cancer. Polymeric nanoparticles (NPs) are powerful, highly tunable carrier systems that may be able to overcome those obstacles. Several studies have shown appropriately-constructed polymeric NPs cross the blood–brain barrier, increase drug bioavailability, reduce systemic toxicity, and selectively target central nervous system cancer cells. While no studies relating to their use in treating brain cancer are in clinical trials, there is mounting preclinical evidence that polymeric NPs could be beneficial for brain tumor therapy. This review includes a variety of polymeric NPs and how their associated composition, surface modifications, and method of delivery impact their capacity to improve brain tumor therapy.

## 1. Introduction

Malignant tumors of the central nervous system (CNS), the vast majority of which originate in the brain [1], are the 13th leading cause of cancer mortality worldwide according to GLOBOCAN 2020 [2]. Glioblastoma (GBM) is derived from astrocytes and accounts for 49% of all malignant CNS tumors, making it the most common form of CNS cancer [3]. Despite decades of work aimed at developing new therapies to target GBM, its prognosis remains poor. Worldwide, the median length of survival for GBM patients is approximately 8 months, and even surgical intervention followed by rapid initiation of radiation and chemotherapy standard-of-care treatment only increases the median survival length to 14 months [1,4] with a five-year survival rate of 5–10% [3]. Similarly, the five-year survival rate for individuals with any form of primary malignant brain tumor is just 20% [3]. The challenges associated with successfully treating brain cancers are numerous. Brain tumors often recur following surgical resection. They also frequently occur in areas that are too difficult or dangerous for gross total resection, necessitating the use of alternative or combination treatment strategies. Additionally, most drugs are incapable of crossing the blood–brain barrier (BBB) and blood–brain tumor barrier (BBTB) in sufficient quantities to halt tumor growth. Furthermore, while stereotactic radiosurgery-based approaches are effective in ablating a variety of brain tumors that are visible on MRI and other neuroimaging modalities, these methods are not as effective in treating tumors with high recurrence rates, including metastatic brain cancers and GBM [5].

Numerous approaches aimed at overcoming these limitations have been developed over the years, with variable success [6]. Some of the most promising techniques in development involve the use of nanocarrier systems to bypass the BBB and BBTB, selectively target brain cancer cells, and release anti-cancer compounds into diseased tissue while limiting toxicity to systemic and healthy brain tissue. Nanoparticles (NPs)—carriers ranging from 10–1000 nm in diameter—can be loaded with chemotherapeutic drugs, nucleic acids, antibodies, and other proteins and peptides. These carriers can be engineered from a variety of materials, with metal, lipid, and polymer-based NPs being the most tested in neurological disease research [7]. The general structure of polymeric NPs consists of a core polymer with therapeutic agents either surface-bound or encapsulated and coated with targeting and/or hydrophilic molecules to increase circulation half-life and specific delivery [8].

In recent years, polymeric NPs developed for CNS tumor treatment have been modified with various moieties capable of interacting with the BBB and tumor cells. Appropriate subtypes can be selected by assessing the polymer pharmacokinetics, modifiability, and payload delivery best suited for any given study. Additionally, polymers can be engineered to form other nanomaterials that may be useful for therapeutic development. For example, a recent review detailed the ability to produce optical nanofibers from polymers that can aid in phototherapy, drug delivery, sensing, and more [9]. Others have explored the potential of novel noninvasive delivery methods, like loading polymeric NPs into neutrophils or monocytes, to enhance their transport to brain tumors [10]. Though many challenges remain, mounting evidence from promising preclinical studies utilizing a variety of approaches suggests polymeric NPs may prove effective in treatment of CNS malignancies. This review provides a summary of the major types of polymeric NPs used in brain cancer studies, the ways in which these NPs can be modified and delivered in order to bypass the BBB, and strategies for specific targeting of NPs to cancer cells.

## 2. Major Polymers in Nanoparticle-Based Brain Cancer Research

### 2.1. Polyanhydride

Polyanhydride is formed by carboxylic acid polymerization with anhydride linkages [11] and is an exceptionally well-characterized and proven biocompatible and biodegradable polymer for cancer therapy. Before discussing the current state of polyanhydride in nanomedicine, it is imperative to address its role in current therapeutic treatment for GBM and other primary brain malignancies. The 1,3-bis(2-chloroethyl)-1-nitrosourea (BCNU) wafer, more commonly known as the Gliadel^®^ wafer, is a biodegradable carmustine-loaded polyanhydride-based implant that was first approved by the US Federal Drug Administration (FDA) in 1996 for patients with recurrent GBM as an adjunct to surgery [12]. In 2003, that approval was expanded to treatment for patients with newly diagnosed high-grade malignant gliomas as an adjunct to surgery and radiation therapy [12].

Gliadel^®^ wafers were the first FDA-approved method of delivering local, sustained-release chemotherapy to brain tumors [13]. In combination with temozolomide and radiotherapy, Gliadel^®^ wafers have become part of the gold standard for treatment of GBM and have significantly improved median patient survival times [14]. A 2022 observational study of 506 malignant glioma patients receiving adjuvant treatment with Gliadel^®^ wafers reported a median overall survival of 18.0 months, with 39.8% and 31.5% of patients surviving two and three years, respectively [15]. Building on this progress, additional approaches utilizing NP-based formulations encapsulating chemotherapeutic drugs and other compounds have been increasingly studied in an attempt to further improve brain tumor treatments and outcomes.

With the success of the Gliadel^®^ wafer, polyanhydride NPs have been investigated as a potential platform for novel brain cancer therapeutics. Polyanhydride NPs are highly tunable and their degradation profile can be modified from days to months, depending on their copolymer composition [16]. Like other polymeric NPs, surface ligands can also be added to polyanhydrides to enhance targeted delivery [17]. Polyanhydrides are highly hydrophobic, which allows their rate of release and erosion to be largely constant and predictable [18]. However, their erosion characteristics may limit their targeting potential due to inadequate ligand retention times [19]. Additionally, polyanhydrides are relatively difficult to synthesize compared to other polymers utilized in nanomedicine and their potential to acylate nucleophiles can result in limited stability of loaded peptides and proteins [20]. Nevertheless, Brenza et al. reported that polyanhydride NPs could successfully cross the BBB via cell-based delivery and transcytosis [21]. While further studies examining polyanhydride NPs as effective carriers for brain tumor therapy have been rather limited, these findings combined with the proven history of Gliadel^®^ wafers suggest polyanhydrides could be an effective platform for nanomedicine.

### 2.2. Poly (lactic-co-glycolic acid)

Poly (lactic-co-glycolic acid) (PLGA) is an FDA and European Medicine Agency (EMA)-approved biodegradable anionic polymer that is widely used to encapsulate chemotherapy drugs, anti-inflammatory drugs, antibiotics, and proteins for the treatment of a variety of conditions [7,22]. In addition to its frequent use in microsphere and microparticle drug delivery systems [23], PLGA is a common platform for NP-based therapies due to the unique properties of PLGA NPs, including their simple biodegradability, relative ease of synthesis, tunability, commercial availability, sustained drug-release properties, and biocompatibility [24]. A recent study by Maksimenko et al. utilizing doxorubicin-loaded PLGA NPs coated with poloxamer 188, which is a copolymer surfactant capable of repairing function in damaged cells [25], reported the carriers could penetrate both the BBB and intracranial tumors to result in significant anti-tumor efficacy in vivo [25]. Another group reported that PLGA NPs loaded with the chemotherapeutic drug morusin and conjugated to chlorotoxin, which is a peptide specific for particular chloride channels expressed in glioma cells, resulted in significant anti-tumor effects against two human glioblastoma cell lines in vitro [26]. Similar in vitro findings using various modified forms of PLGA NPs have been published in recent years [27,28,29], and studies investigating their use against brain tumor-bearing rodent models have also been largely encouraging [30,31]. There is considerable literature documenting the potential of PLGA-based nanocarrier systems to treat CNS tumors via specific surface modifications and loading contents. A general overview of these tunable features for PLGA and other polymeric NPs is shown in Figure 1.

PLGA NPs are degraded into lactate and glycolate, compounds that can be further metabolized through the Krebs cycle [32]. The overall hydrophobicity and degradation rate of PLGA copolymers depends on their ratio of poly (glycolic acid) (PGA), which is hydrophilic, to poly (lactic acid) (PLA), a hydrophobic polymer [32]. Increased hydrophobicity results in a slower rate of degradation and, as a result, a slower rate of drug release [32,33]. PLGA polymers are negatively charged; therefore, cellular uptake through negatively charged cell membranes is limited. However, surface modifications can result in neutral or positively charged PLGA NPs that may penetrate the blood–brain barrier more effectively [7,34]. One challenge in the use of PLGA NPs in therapeutic contexts is their poor drug loading efficiency [34]. Another challenge to the use of PLGA NPs is that high burst release from PLGA NPs, which may be due to drug adsorption to NP surfaces [22], results in low levels of drugs reaching target cells or tissues [34]. Additionally, the production of acids following degradation—a common drawback of biodegradable polymers—can destabilize acid-sensitive drugs and peptides carried in PLGA NPs [35], though there have been many efforts to limit this issue [24,35].

### 2.3. Poly (β-amino ester)

Poly (β-amino ester) (PBAE) is an easily synthesized, biodegradable, and biocompatible cationic polymer commonly used to construct NPs that can deliver polynucleotides and other acid-labile compounds [36]. PBAE NPs are uniquely suited to carrying these types of cargo due to their high efficacy [37] and polyamine nature, which allows these polymers to act as pH buffers [38]. This buffering capacity has been found to enhance the ability of PBAE NPs to escape from endosomes following endocytosis, allowing expression of nucleic acids within target cells [39]. PBAE polymers have an established safety profile [36] and also maintain low cytotoxicity compared to other cationic polymers such as polyethylenimine (PEI) [40] due to their rapid degradation under physiological conditions, which further enhances nucleic acid delivery from NPs to cells [36,37]. Although clinical usage of PBAEs has historically been limited due to their rapid hydrolysis and substantial cationic properties—both of which contribute to PBAE polymer instability in the blood [41]—surface modification studies have attempted to mitigate this disadvantage [42]. Additionally, an extensive library of more than 2300 PBAE polymers was introduced by Anderson et al. in 2003 to determine ideal polymers for targeting particular cell types and tissues [43] and newer libraries have since been developed to expand upon these findings [40,44]. However, while some studies have shown that PBAE NPs can preferentially transfect GBM cells over normal brain tissue in vitro and in vivo [44], others have reported that PBAE and other cationic NPs encounter adhesive interactions with the extracellular matrix (ECM) that limit their ability to achieve widespread gene transfer in brain tumor tissue [45].

### 2.4. Chitosan

Chitosan is a biodegradable polymer created by deacetylation of the widely abundant, naturally occurring polymer chitin [46]. Given its primary and secondary hydroxyl groups, and its amino group, a range of structures can be derived from chitosan by N-linked and O-linked modifications [47]. NPs constructed using chitosan and its derivatives often possess mucoid and cationic properties that facilitate adherence to mucous membranes and sustained drug release [46]. Furthermore, the adherent properties of chitosan allow it to pass transcellularly through the endothelium and epithelium, making it a promising candidate for crossing the BBB [48] and for nose-to-brain delivery [46] via chitosan-based polymeric NPs. While one potential drawback of chitosan-based drug delivery is its low solubility at physiological pH, this may allow for preferential release in a tumor acidic environment or in intracellular endosomes [47].

Chitosan can also be used as a polymer coating and in hybrid NP carriers. In vitro and in vivo studies have demonstrated the potential for these NP drug delivery systems for GBM. Shevtsov et al. reported that hybrid chitosan-dextran superparamagnetic NPs demonstrated enhanced internalization in U87 and C6 glioma cells compared to those coated with dextran alone, and further demonstrated accumulation of these particles in orthotopic C6 gliomas in rats [49]. Successful tumor growth reduction via magnetically guided delivery of folate-grafted, chitosan-coated magnetic NPs containing doxorubicin to human U87 GBM cells in a subcutaneous tumor model in mice has also been demonstrated [50].

### 2.5. Poly(amidoamine) Dendrimers

Poly(amidoamine) (PAMAM) dendrimers are flexible, non-toxic, and biocompatible branched NPs that were first synthesized in 1985 [51]. The structure of PAMAMs consists of an initiator core that anchors dendrimer growth, interior dendrimer layers and branches, and terminal functionalized branches in an outer layer [52] that gives PAMAMs an extremely high surface-area-to-volume ratio [51]. Advantages of PAMAMs in drug delivery applications include their high solubility, stability, small size, and presence of readily modifiable surfaces [53]. While PAMAMs have been associated with cytotoxicity, their cytotoxic effects can be modulated by selective modifications to the terminal functional branches in the outer layer [51].

The cationic nature of PAMAMs, coupled with their large hydrophilic surface area, has made them of particular interest in drug delivery [52]. Sarin et al. demonstrated successful crossing of the BBTB by functionalized dendrimers with diameters of less than 11.7 to 11.9 nm in rodents with orthotopic RG-2 malignant glioma, and further noted the accumulation of dendrimers with long half-lives within glioma cells [54]. Additionally, Moscariello et al. demonstrated that a PAMAM dendrimer bioconjugate with streptavidin adapter was capable of transcytosis across the BBB both in vitro and in vivo [55].

### 2.6. Poly(caprolactone)

Poly(caprolactone) (PCL) is a biodegradable and non-toxic polymer characterized as a semi-crystalline aliphatic polyester that is obtained from a monomer ε-caprolactone ring opening [56]. PCL is a promising, FDA-approved polymer in the development of NP therapies. This is primarily due to its versatility as the combination of PCL with other polymers directly influences its crystallinity, solubility, and rate of degradation, allowing PCL-based drug delivery systems to be utilized for a variety of different approaches [56]. One study highlighting the potential of PCL utilized paclitaxel-loaded PCL NPs conjugated to Angiopep-2, which is a ligand that binds to the low-density lipoprotein receptor related protein (LRP) [57]. LRP is overexpressed on BBB and glioma cells [58], and Xin et al. reported significantly higher penetration, distribution, accumulation, and anti-glioblastoma efficacy in tumor-bearing mice compared to NPs without Angiopep-2 [57].

PCL is a highly stable polymer, due in part to its strong hydrophobicity and crystallinity [59], that requires two to four years for complete degradation [60]. The two stages of PCL degradation include non-enzymatic breakage of ester linkages, followed by enzymatic fragmentation [59]. While this process is slow, hydrophilic polymers can be added to shorten the degradation time [61]. PCL permits modification of its physical, chemical, and ionic properties; therefore, it can be designed to fit the intended properties for specific drug deliveries, ultimately improving therapeutic efficacy. Although most of the current formulations used in drug delivery are satisfied by these PCL modifications, the hydrophobicity of PCL is a drawback that limits its use [59].

### 2.7. Poly(alkyl cyanoacrylate)

Poly(alkyl cyanoacrylate) (PACA) NPs are typically made from alkyl cyanoacrylate monomers and their mixtures through a one-step mini-emulsion process that results in varying levels of PACA particle degradability [62]. Particles with longer alkyl chains typically degrade at a slower rate [63]. PACA NPs have shown significant promise in delivering drugs across the blood–brain barrier, as well as in infiltrating solid tumor structures [64]. The intracellular drug availability provided by PACA delivery is primarily affected by the degradation of its NPs, which occurs through surface erosion and hydrolysis of esters (with or without esterases), allowing for the release of hydrophobic drugs [65]. PACA has been studied in a variety of physicochemical environments and several studies have utilized PACA-based NPs for hydrogels and delivering nucleic acids and peptides in vivo [66]. One such study by Baghirov et al. reported their novel PACA NPs could be transported across the BBB and into brain tissue with ultrasound-mediated delivery [67]. Similarly, Andrieux et al. summarized several different PACA NP-based approaches and concluded they could readily cross the BBB in animal and cell models with appropriate surface modifications [68]. One such modification is the addition of polysorbate 80, a surfactant that may enhance NP delivery via transcytosis across the BBB [69]. Additionally, the biodegradability of PACA allows for continuous drug delivery rather than in bursts, which are present in traditional cancer treatment methods such as chemotherapy [66]. Furthermore, PACA NPs are reportedly capable of overcoming multidrug resistance, which allows tumors to resist chemotherapeutic drugs such as doxorubicin due to P-glycoprotein overexpression [70]. The mechanistic explanation for this involves the formation of an ion pair between PACA degradation products and doxorubicin [68]. This suggests the PACA polymer base could be a suitable choice for chemotherapeutic approaches in the field of neurosurgery.

## 3. General Modifications

### 3.1. Polyethylene Glycol

Polyethylene glycol (PEG) is a hydrophilic polymer that can be covalently attached to NPs and other therapeutics to increase their systemic circulation time [71]. PEG is classified as Generally Regarded as Safe (GRAS) by the FDA and several protein therapeutics coated with PEG, or “PEGylated”, have been FDA approved since 1990 [72]. The conjugation of PEG to NP surfaces reduces their recognition by immune cells through minimizing protein adsorption via steric hindrance, thereby increasing bioavailability [8]. More specifically, PEGylated NPs have been shown to exhibit fewer surface interactions with plasma proteins and cell membranes than non-PEGylated controls, meaning they are more resistant to aggregation, opsonization, and phagocytosis [71,72]. Consistent with its FDA categorization, PEGylation of NPs has not been shown to increase toxicity [7,8] and is one of the most popular modifications used to enhance the effects of nanotherapeutics. A table summarizing the utilization of PEG, other common modifications, and polymer subtypes above can be seen in Table 1.

### 3.2. pH

The extracellular pH in solid tumors is more acidic compared to normal tissue [74]. Normal tissues maintain a pH of ~7.4, whereas tumor microenvironments exhibit a pH of ~6.5 and can thereby be targeted by pH-responsive nanocarrier modifications and triggered drug release [75]. Development of tumor pH-sensitive drug release systems has been studied extensively. Research has shown incorporation of weak acids and other pH-sensitive compounds into polymeric NPs can result in drug carriers that are stable at physiological pH while destabilized and precipitated in the tumor microenvironment, resulting in drug delivery [76]. One group investigating NPs for brain tumor therapy attached H_7_K(R_2_)_2_, a pH responsive peptide, to the surface of PLGA-based, PEGylated NPs in order to enhance targeting of malignant glioma cells in vivo [77]. While H_7_K(R_2_)_2_, remained unexposed under physiological conditions due to hydrophobic interactions between PLGA and the H_7_ residues, the acidic tumor environment protonated the imidazole ring of H_7_, thus making the H_7_K(R_2_)_2_ more hydrophilic and selectively exposing the cell-penetrating ligand to glioma cells [77]. Similar approaches have been used to modify various polymer cores to selectively “activate” NPs in acidic environments [76,78]. Additionally, multiple studies have explored how utilizing pH-responsive nanomaterials can enhance selective targeting of brain tumor cells by polymeric NPs [77]. Laboratories have investigated how other stimuli, such as temperature, redox gradients, enzyme concentration, and magnetic field, can be exploited alongside pH to enhance nanocarrier delivery and release in tumor environments [78]. Taken together, designing polymeric NPs that take advantage of tumor-specific stimuli to enhance targeted delivery and release may prove to be a viable strategy for CNS tumor therapy.

### 3.3. Size

While NPs are 10–1000 nm in diameter by definition, the optimal NP size for clinical brain tumor therapy remains unclear for a variety of reasons. Size plays a key role in NP stability, ability to pass through the BBB/BBTB and spread throughout the brain parenchyma, and likelihood of being endocytosed for cell-mediated delivery [7,71]. Systemically administered NPs with diameters <5 nm are cleared via renal filtration, while larger NPs (>200 nm in diameter) cannot effectively reach the BBB due to splenic sequestration [7]. Fortunately, polymeric NPs can be engineered to precise size specifications [79]. One study of PEG-coated PLGA NPs found that 100 nm particles had longer circulation time and enhanced penetration of the brain parenchyma compared to 200 nm and 800 nm particles in a traumatic brain injury mouse model [80]. Another study utilizing PEG and vitamin E-coated polystyrene NPs of various sizes reported the smallest diameters had the highest brain uptake levels (25 > 50 > 100 > 500 nm) in a rat model [81]. However, conversely, Nowak et al. found spherical polystyrene NPs with diameters of 200 nm crossed the BBB more effectively than 100 and 500 nm spheres in a microfluidic model [82]. Furthermore, other studies have found NP size to have limited or no effect on their ability to penetrate the BBB [83]. These contrasting findings highlight the most common challenges in developing brain tumor therapeutics, as the BBB, BBTB, extracellular space, pore size, and overall physiology varies substantially across in vivo and in vitro models, especially compared to humans [71,82].

NP size also impacts their ability to spread throughout the brain parenchyma, thus affecting therapeutic delivery to tumor regions after crossing the BBB [71]. Similar to BBB studies, accurately replicating the human brain parenchyma and extracellular space remains a significant challenge in bringing NP therapeutics to the clinical trials [71]. Thorne et al. published findings in 2006 suggesting the extracellular space pores in rat brains is up to 64 nm, meaning larger NPs would be unable to further penetrate brain tissue after crossing the BBB [84]. However, Nance et al. later reported in 2012 that larger particles (e.g., up to 114 nm) could still diffuse through the brain extracellular space in both rats and humans with PEG or carboxyl moiety (COOH) coatings [85]. Furthermore, Thorne et al. also suggests that there may be around 25% or more pores in the human brain extracellular space with diameters equal to or larger than 100 nm, with some even exceeding 200 nm [84]. These findings have made clear that more research is required to elucidate the ideal NP size for delivery throughout brain tissue.

### 3.4. Shape

Polymeric NPs can be engineered in a variety of shapes, though spherical particles are the most studied form. Shape has been shown to influence NP pharmacokinetics, cellular uptake, and BBB penetration [86,87]. Spherical NPs may more readily interact with cellular surfaces and thereby have enhanced uptake and clearance by the spleen before reaching tumor regions compared to long, cylindrical NPs, as summarized by Truong et al. [86]. Similarly, Christian et al. reported enhanced circulation time of flexible filamentous micelles (“filomicelles”) compared to spherical micelles in mice [87]. Furthermore, both micelle forms were loaded with paclitaxel and mice treated with filomicelles had enhanced tumor shrinkage and tumor cell apoptosis. Other studies have also reported findings of increased circulation time for various rod-shaped NPs, suggesting that shape is an important characteristic in the development of polymeric NPs for effective brain tumor therapy [88,89]. One particular study investigated endothelial uptake of modified polystyrene nanospheres and nanorods in the brain [90]. Kolhar et al. found the nanorods to have a sevenfold higher accumulation in mouse brains, though they also reported increased accumulation of these particles in the lungs, kidneys, heart, and spleen. Similarly, Nowak et al. reported significantly enhanced transport across a BBB microfluidic model of polystyrene rod-shaped NPs compared to spheres [82]. Indeed, while spherical NPs are most popular, considerable debate regarding the ideal shape for CNS treatment remains. Furthermore, it likely also depends on the mechanism of delivery, as approaches not dependent on crossing the BBB may prefer different conformations than those that must travel through the systemic circulation.

## 4. Receptor Targeting for Blood–Brain Barrier Penetration

Delivering NPs to the brain for tumor therapy remains a significant challenge due to the BBB. In order to cross the BBB, NPs can be engineered to take advantage of transport processes such as adsorptive-mediated transcytosis (AMT), receptor-mediated transcytosis (RMT), and cell-based delivery (described in the Mechanisms of Delivery section below) [91]. AMT involves electrostatic interactions between positively charged ligands and negatively charged brain capillary endothelial cell membranes [92]. While NPs may be able to take advantage of this process, it is unclear if AMT factors significantly into the delivery of endogenous compounds through the BBB [91]. Conversely, targeting endothelial cells present on the BBB for RMT is among the most common approaches to enhance drug delivery to the brain parenchyma [93]. RMT involves receptor-mediated endocytosis on the luminal side of the BBB, followed by trafficking and sorting through endothelial cells, and concluding with the release of contents to the brain parenchyma [94]. Specific targets for RMT include the transferrin receptor (TfR) [95], insulin receptor [96], low density lipoprotein (LDL) receptor (LDLR) [94], melanotransferrin [94], CD98 [97], and various others [7,94,98]. An overview of potential transport pathways for NPs to penetrate the BBB is shown in Figure 2.

Some receptors, like TfR, are also overexpressed on GBM cells and may be suitable targets for enhancing delivery of NPs through both the BBB and ultimately to brain tumor tissue [99]. TfR is the most commonly targeted protein for enhancing delivery of therapeutics through the BBB via RMT [94,95,97]. One reason for this is that TfR is expressed on brain capillary endothelial cells, but not on endothelial cells in other parts of the body [100]. Additionally, there are multiple potential ligands that can be conjugated to NPs for targeting the Tfr, including transferrin (Tf), antibodies, and targeting peptides [101]. Ramalho et al. reported enhanced internalization of TMZ-loaded PLGA NPs coated with monoclonal antibodies for the TfR (OX26 type) in GBM cells through receptor-mediated endocytosis [99]. Another study, conducted by Kuang et al. [102], investigated the use of dendrigraft poly-L-lysine-based NPs conjugated to a peptide capable of targeting TfR on the BBB and glioma cells to deliver RNA and doxorubicin. They reported high tumor targeting efficiency in vitro and increased cellular uptake, slower tumor growth, improved median survival time, and additional anti-tumor effects in vivo.

Various groups have investigated targeting the insulin receptor, which is expressed on BBB endothelial cells, as well [94]. A study led by Shilo et al. demonstrated five times greater brain localization of insulin-targeted gold NPs compared to controls in a mouse model two hours after intravenous injection [103]. Betzer et al. expanded on these findings to show that similar gold NPs coated with insulin could promote NP localization to specific brain regions in mice [104]. Another group found that anti-insulin receptor antibodies covalently attached to human serum albumin NPs resulted in increased delivery across the BBB compared to controls [105], again suggesting the insulin receptor may be a viable target for BBB penetration.

Conjugating ligands capable of binding LDLR to NPs is another potential strategy for enhancing brain delivery [94] as this molecule is expressed on BBB endothelial cells [98]. Apolipoprotein E and B (ApoE and ApoB) can bind these receptors, and various in vivo and in vitro studies have reported increased diffusion of ApoE-coated NPs across the BBB and BBB models, respectively [106,107]. Other studies have reported successful transcytosis through the BBB of therapeutics containing ApoB fragments as well [108]; however, there is limited data associated with the use of these fragments in NP delivery. Nevertheless, others have demonstrated enhanced delivery of NPs bound to peptides with high affinity for LDLR across the BBB and subsequent glioma localization in vivo [109].

Melanotransferrin, or melanoma tumor antigen p97, binds to LDL receptor related protein 1 (LRP1) and is a type of iron-binding transferrin protein [94]. Karkan et al. [110] demonstrated that covalently linking melanotransferrin to chemotherapy drugs, such as paclitaxel and doxorubicin, could dramatically increase drug delivery to gliomas in vivo compared to controls. Another group had similar results with melanotransferrin conjugated to trastuzumab to treat breast cancer brain metastases compared to controls in mice [111]. These findings suggest that melanotransferrin could be yet another protein that may improve NP RMT through the BBB.

CD98 is a transmembrane, glycoprotein heterodimer composed of CD98 heavy chain (CD98hc) and various CD98 light chains [112] and is expressed in various tissues, including endothelial cells on the BBB, where it functions as an amino acid transporter [113]. CD98 is also overexpressed in tumor and inflammatory cells [114] and has been targeted by various NP formulations to amplify their internalization in diseased tissues [112], though studies targeting CD98 for neurodegenerative disease and brain cancer therapy has been limited. One study led by Zuchero et al. investigated the use of bispecific antibodies targeting CD98hc and β-secretase 1 (BACE1) to reduce amyloid beta production in mice and found that levels of anti-CD98hc/BACE1 in the brain were significantly higher than anti-TfR/BACE1 antibodies following intravenous injection [97]. Others have reported on the enhanced expression of CD98 in astrocytic neoplasms [115], suggesting this complex could be a viable target for NP-based brain tumor therapy. However, while targeting CD98 may be a viable way to enhance RMT of NPs and targeting of cancer cells, the potential associated disruption of amino acid transport across the BBB is a concern that must be studied further [94,113].

## 5. Receptor Targeting for Delivery to Brain Cancer Cells

### 5.1. Vascular Endothelial Growth Factor

Vascular endothelial growth factor (VEGF) is a mitogen for endothelial cells and promotes angiogenesis. It is expressed by various cell types, including endothelial cells and several types of tumor cells [116]. While VEGF is a secreted protein, it commonly localizes to cellular membranes and intracellular matrices [117]. Overexpression of VEGF on tumor cells is critical for tumor growth and metastasis through enhanced angiogenesis, and antibodies against VEGF have long been a target for cancer therapies, including many forms of brain cancer [116].

Bevacizumab is a monoclonal antibody against VEGF that has been approved by the FDA for GBM patients whose condition did not improve following treatment with TMZ, a staple chemotherapy drug for treating GBM [31]. Bevacizumab has since been incorporated into several NP-based brain tumor therapy studies, with most focusing on loading NPs with bevacizumab for more effective delivery to tumor cells [31,118]. Some of these studies are promising, including one that reported reduced tumor growth and higher anti-angiogenic effect from bevacizumab-loaded PLGA NPs compared to free drug in mice after 14 days [31]. Others have investigated the potential for conjugating bevacizumab or other VEGF antibodies to NP surfaces for more effective cancer cell targeting [117]. For example, Abakumov et al. used a PEG linker to conjugate VEGF monoclonal antibodies to magnetic NPs for intracranial visualization of glioma cells with MRI in vitro [117]. Similarly, liposomal NPs were conjugated to a novel VEGF monoclonal antibody through a PEG linker and intravenously injected into rats with C6 gliomas by Shein et al. [119]. They reported both specific accumulation of these NPs in tumor tissues and engulfment by glioma cells. Studies utilizing similar NPs loaded with cisplatin and conjugated to antibodies against VEGF and its receptor type II (VEGFR2) have since demonstrated enhanced drug delivery and uptake by glioma cells in vivo [120].

### 5.2. Epidermal Growth Factor Receptor

Epidermal growth factor receptor (EGFR) is one of four types of receptor tyrosine kinases within the ErbB protein family and is overexpressed in many cancers, including in 40–50% of patients with GBM [71,121]. Several therapeutics for cancers throughout the body targeting EGFR and its variants have been developed over the years, including the FDA-approved EGFR tyrosine kinase inhibitors afatinib and dacomitinib, and monoclonal antibodies against EGFR such as cetuximab, panitumumab, and nimotuzumab [122]. However, many of these therapeutics have not yet been shown to be clinically effective against GBM and other brain cancers, due, in part, to challenges in delivery across the BBB [123]. Westphal et al. published a large summary of various EGFR-targeted therapeutics for GBM in 2017, again citing brain delivery as the most common pitfall [124].

NPs conjugated to EGFR-targeting antibodies or loaded with EGFR inhibitors have been increasingly studied in recent years [71] with the aim of enhancing tumor-specific delivery and inhibition. For example, Mortensen et al. conjugated cetuximab to immunoliposomal NPs via PEG linkage and reported enhanced uptake and accumulation in an intracranial U87 MG xenograft mouse model [125]. In another study conducted by Erel-Akbaba et al., solid lipid NPs loaded with siRNA against human EGFR were found to exhibit dose-dependent EGFR knockdown when tested in human U87 glioma cells [126]. Furthermore, they found significantly decreased EGFR mRNA in tumor tissue after retro-orbital injection of solid lipid NPs loaded with EGFR siRNA at 8, 9, and 11 days post-GL261 tumor implantation in mice. Other studies have investigated nanosyringes conjugated to EGFR binding peptides for localized delivery specifically to tumor regions in intracranial xenograft mouse models [71]. Additionally, a Phase I clinical trial using weekly administration of novel nanocellular compounds loaded with doxorubicin and conjugated to panitumumab over 8 weeks following standard therapy (radiation and TMZ) demonstrated no dose limiting toxicity in 14 patients with recurrent GBM [121]. Other NP studies have reported similar findings in various rodent and canine models [71], leading to additional clinical trials investigating EGFR-based NPs. These include a recently-completed Phase I study led by groups from EnGeneIC Limited and Johns Hopkins University to investigate EGFR-targeting 400 nm minicells loaded with doxorubicin in the context of GBM; results currently pending [127].

## 6. Mechanisms of Delivery

### 6.1. Focused Ultrasound

Focused ultrasound (FUS), is a promising technique that uses a minimally invasive approach to treat a variety of diseases, including those related to the brain [128]. FUS has the unique ability to safely and reversibly open the BBB, thus increasing the ability of drugs to reach normally inaccessible brain sites [129]. This finding has been demonstrated in multiple recent clinical trials [8] with no adverse radiological findings at three month follow-up.

The BBB, which is primarily formed by capillary endothelial cells, astrocyte end-feet, and pericytes, serves as a biological barrier that only allows channel-mediated and small molecules to pass through, under normal conditions. While the BBB primarily functions to protect the brain, it also restricts the types of drugs that can reach the brain. In combination with the use of microbubbles, which are similar to red blood cells in size at ~10 μm in diameter [130], FUS improves drug delivery, allowing for more effective passage of drugs across the BBB [131]. In this way, the use of FUS prior to NP dosing results in enhanced delivery of NPs to cancerous cells [71].

Several studies in recent years have investigated the use of FUS for polymeric NP delivery across the BBB. Yang et al. recently reported successful delivery of lipid-polymer hybrid NPs loaded with clustered regularly interspaced short palindromic repeats (CRISPR)-associated protein 9 (CRISPR/Cas9) plasmids targeting a temozolomide drug-resistance gene to glioblastoma cells in vivo [132]. They subsequently found significantly inhibited tumor growth and prolonged survival times in tumor-bearing mice. Separately, Nance et al. reported pressure-dependent delivery of PLGA-based NPs into regions of the brain parenchyma of a rat model where FUS was used to temporarily disrupt the BBB [133]. Other studies have published similar findings of enhanced NP transport across BBB and localization to target regions via FUS [134,135]. An illustration of FUS and other approaches that can be used to deliver NPs to brain tumor tissue is shown in Figure 3.

Through the combined use of FUS to generate oscillation openings and microbubbles—most commonly Optison, Definity, and Sonovue—to transport drugs, medications are more likely to reach their target sites in the brain and have increased efficacy [136]. Despite these potential benefits, however, it is important to note that FUS-mediated drug delivery presents certain risks that must continue to be investigated, including acute complications such as microhemorrhages and vacuolation of pericytes and other cells at the BBB following sonication, likely due to the temporary BBB disruption [129,137].

### 6.2. Convection-Enhanced Delivery

Convection-enhanced delivery (CED) was developed at the National Institutes of Health in the early 1990s as a way to bypass the BBB for localized drug delivery [138]. CED involves stereotactic placement of one or more catheters and an external pressure gradient to pump drugs via fluid convection directly into regions of interest within the brain [138,139]. This method also results in a greater volume of distribution compared to diffusion, thereby allowing drugs to spread further throughout the brain parenchyma [140]. CED may be particularly beneficial for delivering drug-loaded NPs with tumor cell-targeting properties to remaining cancer cells after surgical resection in clinical settings [139]. Indeed, there are multiple ongoing in vivo studies and clinical trials aimed at using CED to deliver various chemotherapeutic drugs to glioma cells [71,141]. However, no trials involving CED have established a definitive increase in glioma survival time compared to the current standard of care [141,142]. Additionally, with CED comes potential downsides, such as risk of infection, catheter obstruction, and limited therapeutic administration windows compared to systemic treatments [71].

One example illustrating the potential of CED in brain tumor nanomedicine investigated the delivery of PBAE NPs complexed with DNA plasmids coding for the luciferase protein (for fluorescent tracking) and p53 tumor suppressor protein to the brain parenchyma in tumor-bearing rats [143]. Mastorakos et al. reported that the PBAE NPs, which were also conjugated to polyethylene glycol to avoid adhesive trapping in the brain, rapidly traversed the brain parenchyma and orthotopic brain tumors. Furthermore, rats treated with these same DNA-loaded NPs displayed effective gene transfection of tumor cells and significantly enhanced survival compared to similar NPs without the polyethylene glycol coating. While these findings are indeed promising, the study did not investigate any potential complications resulting from CED. Another study utilizing similar DNA-loaded NPs administered via CED resulted in transgene expression in both healthy striatal tissue and malignant glioma cells in a rat model [144]. Nevertheless, significantly more replicated studies and research into potential CED delivery of NPs are required before potentially reaching the clinical trial phase.

### 6.3. Nose-to-Brain Delivery

The nose-to-brain technique offers a minimally invasive and relatively convenient option for NP delivery [145]. Drugs delivered nasally to the CNS are directly absorbed via the olfactory and trigeminal pathways, circumventing the blood brain barrier [146], and can exhibit favorable pharmacokinetics and pharmacodynamics [147]. One challenge in nasal delivery of drugs includes the presence of various degrading enzymes in these pathways, including cytochrome P450, as well as mucociliary clearance that reduces drug retention and absorption [148]. While it is known that various therapeutic compounds delivered via nose-to-brain are able to spread from the olfactory and trigeminal pathways to brain parenchyma and CSF [149], the exact mechanism allowing for this is unclear and resulting bioavailability is often low [150].

Studies have shown modified NPs are capable of protecting their contents from enzymatic degradation and clearance via nasal cilia [148]. While most molecules are not likely to reach therapeutic doses in the CNS via intranasal delivery, polymeric NPs including chitosan, PLGA, and PLA have been studied for potential delivery via this method [151]. Craparo et al. demonstrated that mPEG-PLGA labeled with Rhodamine-B were suitable for drug delivery via the nose-to-brain route [151]. Separately, Chu et al. successfully targeted glioblastoma in a rodent model using the nose-to-brain technique to deliver modified TMZ-loaded PLGA particles [152]. In addition to the significant glioma cell cytotoxicity found in rats treated with nasal delivery of these modified NPs, fluorescence imaging showed lower buildup of NPs in other organs and higher brain distribution compared to the same carriers delivered intravenously after four hours [152]. Another study reported prolonged life and decreased tumor growth in tumor-bearing rats after nose-to-brain delivery of modified PCL nano-micelles containing therapeutic compounds [153]. Kanazawa et al. followed up on these findings by attaching a peptide that binds specifically to gastrin-releasing peptide receptor (GRPR) to similar PCL nano-micelles in a subsequent study [149]. GRPR is upregulated in various tumor cells, including glioblastoma [154], and the researchers found selective cellular uptake of and cytotoxicity in C6 glioma cells treated with these PCL nano-micelles. It remains to be seen if these promising preliminary rodent findings and others will be translated to successful human clinical trials, but nose-to-brain delivery may indeed serve as a powerful technique for minimally invasive delivery of brain tumor therapeutics.

### 6.4. Intracranial Hydrogel Delivery

Hydrogels are three-dimensional, hydrophilic gel polymer networks that provide the benefits of minimal toxicity, localized and stimuli-responsive drug delivery, and passively controlled drug release [155]. These same potential advantages can be found in NP-based therapeutic systems, and, indeed, many studies have investigated the use of NP-loaded hydrogel hybrids as a means for managing brain tumors [156]. However, one common obstacle in the use of hydrogels for this purpose is that most chemotherapeutic drugs are hydrophobic, making them largely incompatible with the hydrophilic nature of the gels [157]. Polymeric NPs are highly tunable and can be constructed with hydrophilic shells loaded with hydrophobic drugs, potentially circumventing this issue [7]. NPs can be incorporated into hydrogels in a variety of ways before and after the gelation process, or even by being incorporated into the gel matrix itself as many gel matrices have been shown to be stable environments for NPs [155].

The combined use of NPs and hydrogels has emerged as a potentially powerful tool for improving brain tumor therapy through safe and localized delivery of hydrophobic drugs. Using this combination strategy, drugs can be injected intratumorally or implanted following surgical resection [158]. Still, both mechanisms of delivery are invasive, prompting additional studies aimed at developing nanoscale hydrogels—so-called “nanogels”— for intravenous approaches [159]. These nanogels have the potential to add NP-based advantages, such as the ability to easily cross the BBB, be internalized by cells, and efficiently encapsulate drugs, to hydrogel systems [160]. Combining these with hydrogel swelling capabilities, or ability to expand in aqueous solution [161], and hydrophilicity makes nanogels a powerful tool for drug delivery [158]. Many recent in vivo studies have confirmed the ability of various nanogels to cross the BBB and selectively target tumor cells [162,163,164]; however, further investigation is required to determine whether these systems can be safely and effectively incorporated into clinical practice.

### 6.5. Cell-Based Delivery

NPs can be internalized by monocytes, neutrophils, and stem cells for delivery to brain tumor tissue. The utilization of immune cells such as monocytes and neutrophils as vectors for NP delivery is especially appealing as these cells readily cross the BBB to sites of injury, inflammation, and tumor growth [10]. This cell-based approach potentially circumvents the need for NPs to traverse the BBB and brain parenchyma themselves. Additionally, among the greatest challenges of delivering systemically-injected NPs to brain tumor sites remains their rapid uptake and removal from circulation by the reticuloendothelial system, consisting of monocytes/macrophages in the liver, spleen, and other fixed tissues [7,71]. In theory, monocytes/macrophages and neutrophils could be obtained from patients and loaded with drug-containing NPs in vitro before reintroducing them to the bloodstream via intravenous injection, allowing them to eventually migrate into and around tumors [165].

The potential for cell-based delivery of NPs in brain tumor therapy has been increasingly studied in recent years. In 2020, Ibarra et al. reported the efficient delivery of monocytes loaded with conjugated polymer NPs to GBM spheroids and an orthotopic mouse model [166]. In another study investigating neutrophil-based delivery, Wu et al. reported that certain neutrophils internalized with magnetic mesoporous silica NPs containing doxorubicin could actively target inflamed brain tumor tissue following intravenous injection after surgical resection in a mouse model [75]. They noted no effect on neutrophil viability after NP loading and tumor-specific cytotoxicity resulting from NPs accumulating in extracellular traps from neutrophils (NETs) that were then internalized by glioma cells. Another study by Xue et al. had similar findings with liposomes, noting that intravenously-injected neutrophils containing paclitaxel-loaded liposomes could accumulate in the brain and suppress glioma cell growth in mice following surgical resection [167].

Mesenchymal stem cells (MSCs) have also been increasingly studied as potential vectors for glioma treatment, as they are hypoimmunogenic and known to migrate toward tumor cells [168]. In a proof-of-concept study, Roger et al. demonstrated that PLA NPs and lipid nanocapsules could be internalized into MSCs without affecting cell viability and differentiation [169]. In the same study, they also showed that those MSCs could migrate towards human glioma cells in a mouse model following intra-arterial and intracranial injection. Similarly, Clavreul et al. reported that a particular subpopulation of MSCs containing lipid nanocapsules with an organometallic complex injected into the striatum resulted in increased survival time in U87MG-bearing mice compared to mice receiving striatum injections of the organometallic complex-containing lipid nanocapsules alone [170]. Building upon these findings, a 2018 study found injection of MSCs loaded with paclitaxel-containing PLGA NPs into the brains of an orthotopic glioma rat model resulted in paclitaxel resulted in significantly longer survival than rats injected with paclitaxel-primed MSCs or paclitaxel-containing PLGA NPs alone [168]. Furthermore, the same group demonstrated that the same MSCs could transfer paclitaxel to glioma cells and induce tumor cell death in vitro. While there are a limited number of studies successfully showing cell-based delivery of NPs for brain tumor treatment, this approach may be able to alleviate many of the complications in delivering NPs to brain tumor tissue in clinical practice and requires further investigation. A comparison of the various mechanisms of delivery discussed here can be found in Table 2.

## 7. Discussion and Future Directions

The increasing number of encouraging findings from preclinical studies suggests polymeric NPs may ultimately serve as one of the next great innovations in clinical CNS cancer therapy. Nevertheless, questions regarding their ideal size, surface characteristics, loaded contents, and mechanism of delivery remain, which continues to limit their translation to clinical practice. NP-based systems continue to encounter roadblocks such as low delivery to brain tissue due to mononuclear phagocyte-mediated clearance, despite the widespread adoption of modifications like PEGylation to reduce this barrier [94]. Finding an ideal balance between NP stability and degradation while ensuring the contents are released specifically to target regions also remains elusive. Standardization of NP synthesis and development of more representative preclinical models is essential in progressing towards clinical trials and more comprehensive strategies to mitigate this have been explored [176].

The most effective NP formulations for treating CNS malignancies may include hybrid systems, such as those combining gold NPs coated with PEG for improved magnetic resonance imaging of brain tumor periphery [7]. Similarly, NP formulations containing multiple therapeutic components may be beneficial in overcoming intra-tumor heterogeneity in metastases such as GBM [71]. An example of this was shown by Xu et al., who reported that paclitaxel and temozolomide co-loaded PLGA-based NPs resulted in greater inhibition against two glioma cell lines in vivo compared to either single drug [177]. Furthermore, polymeric NPs designed with dual targeting of the BBB and tumor cells may result in greater and more specific delivery to target sites throughout the brain parenchyma. Many moieties overexpressed on both of these were previously described above. This may, thereby, allow for polymeric NPs that could be further modified to target tumor microenvironmental stimuli, such as pH, while maintaining a simpler composition.

Intravenous injection is the least invasive method for NP-based therapies and minimizes risk of complications such as infection and compromised BBB integrity. However, concerns of limited delivery across the BBB and BBTB, as well as systemic toxicity, remain. Much of the recent CNS-targeted NP literature investigates alternative delivery mechanisms like nose-to-brain, FUS, CED, hydrogels or nanogels, and cell-based delivery to overcome these challenges. Of course, these approaches must continue to be studied to ensure their safe use and effectiveness in clinical practice. Nevertheless, these delivery techniques are proving to be a powerful tool for nanotherapeutics. Bringing an effective polymeric NP-based therapy to CNS malignancies will require carefully constructed polymers loaded with biotherapeutics that can selectively target tumor cells while minimizing toxicity to healthy brain tissue and other organs.

## 8. Conclusions

While numerous studies utilizing polymer-based NPs for brain tumor therapy have reported promising findings, few approaches have made it to clinical trial, and none have proven to be effective replacements or additions to the current therapeutic regimens. Pitfalls, including low therapeutic delivery and inconsistent study results, must be overcome if polymeric NP-based therapeutics are to be leveraged in the treatment of brain cancer. While PLGA is the most widely-used polymeric platform in studies of nanocarrier application to the treatment of neurological disease, there is no consensus on the most effective polymer for use in brain cancer treatment, as their variable composition influences the specific targeting ligands, contents, mechanisms of delivery, tumor characteristics, and other factors selected for each study. Furthermore, the ideal NP size and shape is dependent on those same factors and similar studies have reported variable findings in BBB and tumor localization. The differences in physiology and permeability in various in vivo and in vitro models compared to humans has been documented and likely also contributes to the challenges in standardization of effective polymeric NP characteristics. While further investigation is warranted, polymeric NPs remain a promising therapeutic method. Ongoing clinical trials and future studies will hopefully shed more light on the optimal polymeric NP composition for managing brain cancer.

## Figures and Tables

**Figure 1 polymers-14-02963-f001:**
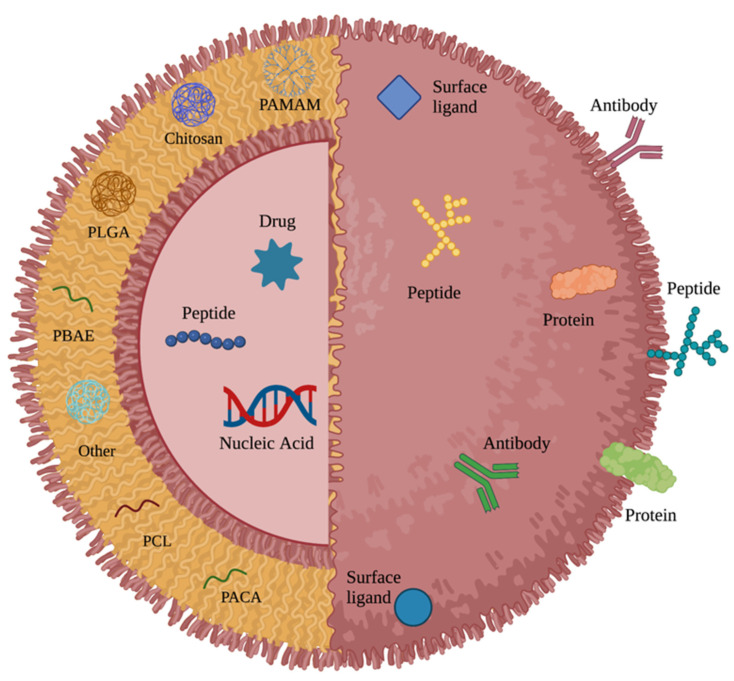
General structure of a polymeric nanoparticle.

**Figure 2 polymers-14-02963-f002:**
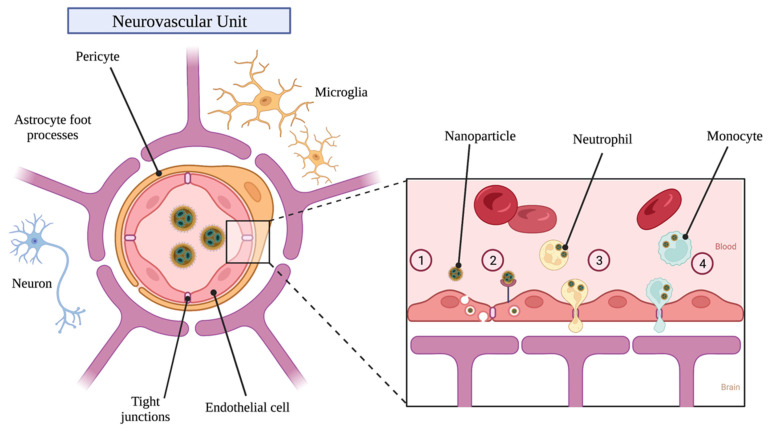
Nanoparticle transport across the blood–brain barrier. (1) Adsorptive-Mediated Transport. (2) Receptor-Mediated Transport. (3) Neutrophil-Mediated Delivery. (4) Monocyte-Mediated Delivery.

**Figure 3 polymers-14-02963-f003:**
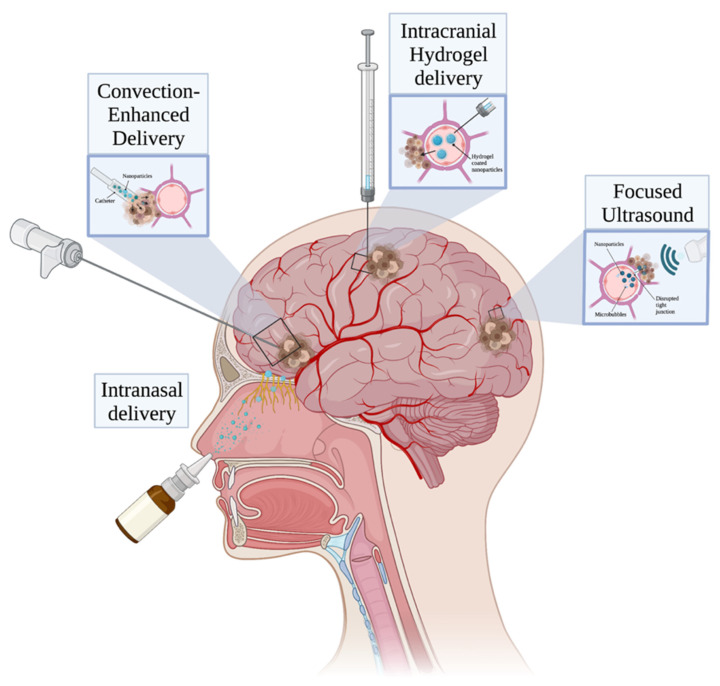
Methods of nanoparticle administration.

**Table 1 polymers-14-02963-t001:** Comparison of polymer subtypes and modifications.

Polymer Type	Common Synthesis Techniques	Advantages	Disadvantages	Specific Uses Cited
Polyanhydride	Most often via polycondensations from diacids or diacyl anhydrides; can also be prepared via solvent evaporation from emulsion, or thiol-ene ‘click’ polymerization, or melt condensation; NP synthesis via nanoprecipitation	Well-characterized; biocompatible; biodegradable; modifiable (depending on copolymer and surface ligand composition); hydrophobic; predictable rate of release/erosion	Rapid erosion can lead to inadequate ligand retention times; difficult to synthesize; limited stability of loaded peptides and proteins due to nucleophile acylation	Gliadel^®^ (BCNU) wafer for local, sustained-release chemotherapy [14]; drug delivery across the BBB [21]; delivery of non-proteinaceous cargo [16]
Poly (lactic-co-glycolic acid)	Co-polymerization of cyclic dimers of glycolic acid and lactic acid; NP synthesis via emulsification-evaporation, nanoprecipitation, phase-inversion, and solvent diffusion; emulsification-evaporation and nanoprecipitation are most commonly used when loading hydrophobic moieties	Widely used; biocompatible; simple biodegradability; easily synthesized; modifiable charge, hydrophobicity, and degradation rate; sustained drug-release; good BBB/tumor penetration	Poor drug loading efficiency; poor drug target delivery efficiency due to high burst release; destabilization of acid-sensitive drugs/peptides	Encapsulation of chemotherapeutics with toxicity profiles indicating sustained, low dosing [7]; microsphere and microparticle drug delivery systems [23]
Poly (β-amino ester)	Conjugate addition of amines to bis(acrylamides) and copolymerization; NP synthesis via solvent/anti-solvent formulation	Established safety profile; biocompatible; biodegradable; easily synthesized; high efficacy; pH buffering capacity; able to escape endosomes and allow intracellular expression of nucleic acids	Instability in blood (rapid hydrolysis) without surface modifications; limited ability to achieve widespread gene transfer due to adhesive interactions with ECM	Delivery of polynucleotides and other acid-labile compounds [36]; delivery of nucleic acids to cells [44]
Chitosan	Enzymatic or chemical deacetylation of chitin, usually through hydrolysis, produces chitosan; NP synthesis via emulsification and crosslinking, microemulsion, precipitation, or ionic gelation	Biodegradable; capable of mucous membrane adherence and transcytosis; sustained drug release; putative preferential release in tumor acidic environment	Low solubility at physiological pH; tendency to aggregate	Nose-to-brain delivery (via mucous membrane adherence) [46]; in situ gelation [73]; tumor targeting via differential pH [47]
Poly(amidoamine) dendrimers	Convergent (beginning with exterior and adding end groups while working towards the core) or divergent synthesis (beginning with core and adding end groups towards the exterior); end group additions via conjugate addition	Biocompatible; flexible, non-toxic; stable; highly soluble; small; modifiable; large hydrophilic surface area; presence of cavities; resistance to denaturation after freezing/thawing	Associated with (modifiable) cytotoxicity; synthesis can lead to heterogeneous mixture of dendrimers unless additional purification steps are completed	Precision-targeting [52]; delivery across the BBB [54]; encapsulating particularly insoluble contents [53]
Poly(caprolactone)	Polycondensation of 6-hydroxyhexanoic acid, or ring-opening polymerization of ε-caprolactone; NP synthesis via nanoemulsification, supercritical fluid extraction of emulsion, or solvent evaporation	Biodegradable; non-toxic; modifiable; stable	High hydrophobicity (slow degradation rate of months/years)	Combination with other copolymers to tailor NP suitability to cargo [56]
Poly(alkyl cyanoacrylate)	Free radical, anionic, and zwitterionic polymerization; NP synthesis via polymerization in aqueous acidic phase or through interfacial emulsion polymerization	Biodegradable; modifiable; enhanced intracellular penetration; capable of overcoming multidrug resistance	BBB translocation ability remains controversial	Hydrogel-incorporated drug delivery [66]; delivery of nucleic acids and peptides [66]; continuous drug delivery (vs. bursts) [66]; instances of multidrug resistance [70]
**Polymer Modification**		**Advantages**	**Disadvantages**	**General Uses**
Polyethylene glycol		Widely used; classified as GRAS; increases systemic circulation time of NPs; reduces recognition of NPs by immune cells; decreases NP aggregation, opsonization, and phagocytosis	Reduced cellular uptake of PEGylated NPs	Modify NP to reduce immunogenicity
pH		Can improve selective tumor targeting via triggered drug release	Limits the types of cargo able to be carried within the NP	Modify NP to selectively target tumor tissue and spare surrounding parenchyma
Size		Can increase NP stability; can potentially increase BBB/BBTB penetration and brain parenchymal spread	Conflicting in vitro/in vivo results on ideal size of NPs for BBB/BBTB penetrance, brain tissue spread, and cellular uptake	Modify NP to increase intra-tumoral spread
Shape		Can modulate NP circulation time, cellular uptake, and BBB penetration	Certain shapes promote accumulation in non-target organs; ideal shape, depending on delivery mechanism, requires further investigation	Modify NP to maximize efficacy based on delivery mechanism (e.g., nose-to-brain vs. across BBB)

**Table 2 polymers-14-02963-t002:** Comparison of polymeric NP mechanisms of delivery.

Mechanism of Delivery	Type(s) of Polymeric NPs Used	Advantages	Limitations
Focused Ultrasound	PLGA [133]	Can reversibly open BBB; targeted delivery; safety supported via clinical trials; minimal systemic effects	Acute complications such as microhemorrhages reported; invasive
Convection-Enhanced Delivery	PLGA [171], PBAE [143], Chitosan [172], PAMAM [173], PCL [174]	High volume of distribution reported; targeted delivery; multiple ongoing clinical trials; potential for use post-resection; minimal systemic effects	No definitive increase in glioma patient survival time reported; infection; limited therapeutic administration windows; invasive
Nose-to-Brain Delivery	PLGA [151], Chitosan [46], PCL [153]	Minimally invasive; easier to study in vivo; bypasses BBB; minimal systemic effects	Exact delivery mechanism and clearance pathways unclear; non-targeted delivery; bioavailability can be low compared to other delivery mechanisms; limited NP clinical studies
Intracranial Hydrogel Delivery	PLGA [155], Chitosan [73], PCL [175]	Potential for use post-resection; targeted delivery; passively controlled drug release; variety of potential approaches; minimal systemic effects	Difficult to use with hydrophobic NPs; invasive; non-targeted delivery
Cell-Based Delivery	PLGA [168]	Minimally invasive; limited clearance via reticuloendothelial system compared to other systemic delivery approaches	Limited NP clinical studies; non-targeted delivery

## Data Availability

Not applicable.

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
