# Peer review of "Polymeric Nanoparticles in Brain Cancer Therapy: A Review of Current Approaches"

_polymers, 2022, doi:10.3390/polym14142963_

Round 1
Reviewer 1 Report
The current manuscript provides a usual account of polymeric nanoparticles for brain cancer intervention. The Authors emphasised on recent approaches in the title. I do not find the manuscript very interesting as it failed to provide a critical outlook and future overview (where to from here). The current approaches include high powered imaging and scanning, targeting and tagging the nanoparticles with small molecules to achieve customised solutions. Mentioning the polymers and their modifications without critical insights on their specific use and advantages serve little purpose. The length of the manuscript is also very short looking at the number of references (222). Try removing old references and keep the most recent ones.
Author Response
The current manuscript provides a usual account of polymeric nanoparticles for brain cancer intervention. The Authors emphasised on recent approaches in the title.
- I do not find the manuscript very interesting as it failed to provide a critical outlook and future overview (where to from here).
Response: The section “7. Discussion and Future Directions” has now been added. This section revisits many of the polymeric modifications and strategies discussed in the review, as well as commentary on additional approaches such as hybrid nanoparticle systems. Also addressed are remaining challenges that may be overcome, in part or in whole, through continued investigation of newer mechanisms of delivery.
- The current approaches include high powered imaging and scanning, targeting and tagging the nanoparticles with small molecules to achieve customised solutions. Mentioning the polymers and their modifications without critical insights on their specific use and advantages serve little purpose.
Response: A table summarizing the major polymer subtypes and modifications has been included with a column on specific uses, in addition to the advantages and disadvantages of each. We hope that this, in addition to the “7. Discussion and Future Directions” section, will sufficiently inform readers of how to formulate polymeric NP systems for specific research applications.
- The length of the manuscript is also very short looking at the number of references (222). Try removing old references and keep the most recent ones.
Response: While a few references were added to accompany the newly-written sections, approximately 50 others have been removed. The total number of references is now 177.
Reviewer 2 Report
In this paper, the authors present a variety of polymeric NPs and how their associated composition, surface modifications, and method of delivery impact their capacity to improve brain tumor therapy. This review indicates while numerous studies utilizing polymer-based NPs for brain tumor therapy have reported promising findings, few approaches have made it to clinical trials and none have proven to be effective replacements or additions to the current therapeutic regimens. Pitfalls, including low therapeutic delivery and inconsistent study results, must be overcome
if polymeric NP-based therapeutics are to be leveraged in the treatment of brain cancer. This review is clear, concise, and suitable for the scope of the journal. Several small suggestions are supplied:
1. Suggest adding more graphs to make it more attractive.
2. Suggest a supply table to conclude some pieces of literature with years.
3. Suggest enhancing the introduction part with some latest reviews, such as:
Biocompatible and Biodegradable Polymer Optical Fiber for Biomedical Application: A Review, Biosensors 11(12):472,2021.
Author Response
In this paper, the authors present a variety of polymeric NPs and how their associated composition, surface modifications, and method of delivery impact their capacity to improve brain tumor therapy. This review indicates while numerous studies utilizing polymer-based NPs for brain tumor therapy have reported promising findings, few approaches have made it to clinical trials and none have proven to be effective replacements or additions to the current therapeutic regimens. Pitfalls, including low therapeutic delivery and inconsistent study results, must be overcome if polymeric NP-based therapeutics are to be leveraged in the treatment of brain cancer. This review is clear, concise, and suitable for the scope of the journal. Several small suggestions are supplied:
Suggest adding more graphs to make it more attractive.
Response: Table 1 has now been included and provides a summary of the major polymer subtypes and modifications discussed in this review. In this table, synthesis techniques, advantages/disadvantages, and specific uses have also been listed to give our review a way to easily compare and contrast these topics between specific polymers and modifications.
Suggest a supply table to conclude some pieces of literature with years.
Response: We hope that the inclusion of the two new tables (Table 1 and Table 2) will address the Reviewer’s concern.
Suggest enhancing the introduction part with some latest reviews, such as:
Biocompatible and Biodegradable Polymer Optical Fiber for Biomedical Application: A Review, Biosensors 11(12):472,2021.
Response: An additional paragraph has been added to the introduction to include recent developments and applications specific to polymers in nanomedicine. Included in this paragraph is a discussion of the suggested review, as well as an overview of some of the recent approaches discussed later in the manuscript such as cell-based delivery.
Reviewer 3 Report
The manuscript titled “Polymeric Nanoparticles in Brain Cancer Therapy: A Review of Current Approaches” presents a summary of relevant studies in the field of nanoparticle (NP) research for the development of effective brain cancer therapies. The authors describe some of the main base materials for NP design, as well as key modifications that can be incorporated to enhance NP therapeutic effect. Moreover, different types of NP delivery mechanisms are discussed.
To improve the quality of the manuscript, the following major revisions are suggested:
1. Section 2 of the manuscript describes some polymeric materials employed for nanoparticle development. Nonetheless, information regarding nanoparticle synthesis is missing, and this element would add significant value to the review. Thus, it is strongly suggested that the authors include a summary table with information concerning NP synthesis methods, delivered drugs/bioactive factors, and observed therapeutic effects.
2. Following the previous comment, the reader would also greatly benefit from having a summary table with a comparison of NP delivery mechanisms. In such table, the authors could include advantages and limitations of each mechanism, as well as the type of nanoparticle they have been used for.
3. Sometimes, the authors discuss polymeric materials in the context of being used as coatings, instead of the main nanoparticle component, as it is observed in the chitosan section, for example. It would be highly beneficial for the reader to present a clear distinction between these two different uses (coating vs NP core component).
Author Response
The manuscript titled “Polymeric Nanoparticles in Brain Cancer Therapy: A Review of Current Approaches” presents a summary of relevant studies in the field of nanoparticle (NP) research for the development of effective brain cancer therapies. The authors describe some of the main base materials for NP design, as well as key modifications that can be incorporated to enhance NP therapeutic effect. Moreover, different types of NP delivery mechanisms are discussed.
To improve the quality of the manuscript, the following major revisions are suggested:
- Section 2 of the manuscript describes some polymeric materials employed for nanoparticle development. Nonetheless, information regarding nanoparticle synthesis is missing, and this element would add significant value to the review. Thus, it is strongly suggested that the authors include a summary table with information concerning NP synthesis methods, delivered drugs/bioactive factors, and observed therapeutic effects.
Response: Table 1 has now been added that includes synthesis techniques, advantages and disadvantages, and specific uses (e.g. chemotherapy delivery, use in hydrogels, etc.).
- Following the previous comment, the reader would also greatly benefit from having a summary table with a comparison of NP delivery mechanisms. In such table, the authors could include advantages and limitations of each mechanism, as well as the type of nanoparticle they have been used for.
Response: Table 2 discusses the advantages and limitations of each discussed mechanism of delivery, as well as the type of nanoparticle they have been used for.
- Sometimes, the authors discuss polymeric materials in the context of being used as coatings, instead of the main nanoparticle component, as it is observed in the chitosan section, for example. It would be highly beneficial for the reader to present a clear distinction between these two different uses (coating vs NP core component).
Response: Chitosan is also commonly used as a NP coating and the distinction between that and core element was not initially clear. The wording has been corrected to more accurately depict this.
Round 2
Reviewer 1 Report
No further comments.
Reviewer 3 Report
The Authors have reasonably addressed the comments. Thank you.